# Generalized Information-theoretic Multi-view Clustering

**Weitian Huang**
School of Computer Science and Engineering
South China University of Technology
Guangzhou, 510006, China

**Sirui Yang**
School of Computer Science and Engineering
South China University of Technology
Guangzhou, 510006, China

**Hongmin Cai**[*]
School of Computer Science and Engineering
South China University of Technology
Guangzhou, 510006, China

## Abstract

In an era of more diverse data modalities, multi-view clustering has become a fundamental tool for comprehensive data analysis and exploration. However, existing multi-view unsupervised learning methods often rely on strict assumptions on semantic consistency among samples. In this paper, we reformulate the multi-view clustering problem from an information-theoretic perspective and propose a general theoretical model. In particular, we define three desiderata under multi-view unsupervised learning in terms of mutual information, namely, *comprehensiveness*, *concentration*, and *cross-diversity*. The multi-view variational lower bound is then obtained by approximating the samples' high-dimensional mutual information. The Kullback–Leibler divergence is utilized to deduce sample assignments. Ultimately the information-based multi-view clustering model leverages deep neural networks and Stochastic Gradient Variational Bayes to achieve representation learning and clustering simultaneously. Extensive experiments on both synthetic and real datasets with wide types demonstrate that the proposed method exhibits more stable and superior clustering performance than state-of-the-art algorithms.

## 1 Introduction

Multiple views of data are often collected by different measurement methods to represent various properties of objects in many practical applications. For example, for images or videos, various visual features such as HOG [1], SIFT [2], and LBP [3] can be acquired by traditional filters and form a kind of multi-view data. These features can be collected from different domains or generated from various sensors. In general, different views can complement each other due to limitations or biases in measurement methods that would result in insufficient information contained in individual views [4].

The ever-increasing network activity brings massive unlabeled data, which makes clustering, the traditional task of machine learning, one of the hot topics in unsupervised learning. Importantly, in clustering task, learning the comprehensive and discriminative representations is crucial for partitioning the samples. Compared to single-view clustering, multi-view clustering (MVC) is widely studied as it exploits complementary information between different views to improve the ultimate performance. Most multi-view clustering algorithms learn the consistent representation among multiple views to realize clustering. The benchmark techniques include view co-regularization [5],

---

[*]Hongmin Cai is the Corresponding Author, whose E-mail is: hmcai@scut.edu.cn.

37th Conference on Neural Information Processing Systems (NeurIPS 2023).

Canonical Correlation Analysis (CCA) [6, 7] and low-rank tensor decomposition [8]. Although different methods have achieved superior clustering performance, they lack a unified theoretical framework to help us further understand multi-view clustering.

Recently, Information Bottleneck (IB) [9] have attracted increasing attention in deep neural network research. The information bottleneck is an information theory principle that helps explain the puzzling success of today's artificial intelligence algorithms. This motivates us to explain multi-view learning using information bottleneck theory. In this paper, we extend from supervised information bottleneck theory to unsupervised learning, and construct a general information-theoretic multi-view clustering framework. The main contributions of our work are illustrated as follows,

- In unsupervised scheme, we define three requirements for multi-view representation learning from the view of information theory, *comprehensiveness*, *concentration*, and *cross-diversity*. Consequently, a general information-based multi-view clustering model (IMC) is proposed to achieve representation learning and clustering simultaneously.

- We present a multi-view VAE scheme to solve IMC by leveraging deep neural networks and Stochastic Gradient Variational Bayes (SGVB). Specifically, the multi-view mutual information is approximated by a variational lower bound, which can be expressed as a unified multi-view representation obtained through multiple variational autoencoders and sampling, followed by a self-training strategy for clustering.

- Experimental results show that the proposed method is effective in learning informative yet minimal representations and exhibits satisfactory generalization ability.

## 2 Variants of Information Bottleneck

### 2.1 Information bottleneck

The information bottleneck principle [9] aims to obtain the most compressed representation of data while retaining task-relevant information. To achieve this, for given input data $\mathbf{x}$ with label $\boldsymbol{y}$, the corresponding codeword $\mathbf{z}$ decided by $p(\mathbf{z}|\mathbf{x})$ can be obtained by optimizing the objective of the information bottleneck,

$$\min_{p(\mathbf{z}|\mathbf{x}):I(\mathbf{Z};\boldsymbol{Y})\geq I^*} I(\mathbf{Z};\mathbf{X}) = \max_{\mathbf{Z}} I(\mathbf{Z};\boldsymbol{Y}) - \beta I(\mathbf{Z};\mathbf{X}) := \mathcal{L}_{IB}. \tag{1}$$

where $I(\mathbf{Z};\mathbf{X})$ represents the mutual information between two random variables $\mathbf{X}$ and $\mathbf{Z}$, which implicitly measures the level of compression in the representation $\mathbf{Z}$. $\mathcal{L}_{IB}$ is to identify the optimal encoding $p(\mathbf{z}|\mathbf{x})$ that minimizes $I(\mathbf{Z};\mathbf{X})$ while ensuring $I(\mathbf{Z};\boldsymbol{Y})$ exceeds a minimal threshold value $I^*$. This constrained problem can then be transformed into a non-constraint problem by incorporating a trade-off factor $\beta$ that balances $I(\mathbf{Z};\boldsymbol{Y})$ and $I(\mathbf{Z};\mathbf{X})$. Such scheme yields sufficient and minimal representations that maximize task-relevant information while minimizing task-irrelevant information, thus balancing prediction and generalization performance.

### 2.2 Unsupervised information bottleneck

When the sample labels are unavailable, our aim is to compress each sample's description to the greatest extent possible while simultaneously maximizing the retention of the principal component information of the data. The unsupervised information bottleneck objective is described in [10],

$$\max_{\mathbf{Z}} I(\mathbf{Z};\mathbf{X}) - \beta I(\mathbf{Z};i) \overset{(a)}{\geq} \mathbb{E}_{p(\mathbf{x})}\left[\mathbb{E}_{p(\mathbf{z}|\mathbf{x})}\left[\log q(\mathbf{x}|\mathbf{z})\right] - \beta D_{KL}(p(\mathbf{z}|\mathbf{x})||q(\mathbf{z}))\right] := \mathcal{L}_{UIB}. \tag{2}$$

where $(a)$ forms the lower bound of $\mathcal{L}_{UIB}$, which is achieved by substituting the intractable posterior $p(\mathbf{x}|\mathbf{z})$ and marginal $p(\mathbf{z})$ with respective variational posterior $q(\mathbf{x}|\mathbf{z})$ and marginal $q(\mathbf{z})$. Note that the lower bound is essentially identical to $\beta$VAE [11]. The detailed derivation process is shown in Appendix A.

In clustering tasks, one aims to discover the underlying structure of data and divide it into distinct clusters. The current deep clustering methods firstly extract deep low-dimensional embeddings and then apply standard clustering techniques. Theoretically, the deep clustering method is expected to

follow the information bottleneck principle while preserving the original data clustering structure. Therefore, one can tailor Eq. (2) to realize deep clustering,

$$\max_{\mathbf{Z}} I(\mathbf{Z};\mathbf{X}) - \beta I(\mathbf{Z};i)$$

$$\geq \mathbb{E}_{p(\mathbf{x})}\left[\mathbb{E}_{p(\mathbf{z}|\mathbf{x})}\left[\log q(\mathbf{x}|\mathbf{z})\right] - \beta D_{KL}(p(\mathbf{z}|\mathbf{x})||q(\mathbf{z})) - \gamma \mathbb{E}_{p(\mathbf{z}|\mathbf{x})}[D_{KL}(p(\boldsymbol{c}|\mathbf{x})||q(\boldsymbol{c}|\mathbf{z}))]\right]$$

$$:=\mathcal{L}_{IBC} = \mathcal{L}_{UIB} - \gamma \mathbb{E}_{p(\mathbf{x})}\left[\mathbb{E}_{p(\mathbf{z}|\mathbf{x})}[D_{KL}(p(\boldsymbol{c}|\mathbf{x})||q(\boldsymbol{c}|\mathbf{z}))]\right]. \qquad (3)$$

Here we introduce a discrete variable $\boldsymbol{c}$ to represent the cluster and denote $p(\boldsymbol{c}|\mathbf{x})$ as the cluster distribution implied by the data. We aim for the cluster inferred by embedding $\mathbf{z}$ to be close to the original cluster distribution. To measure this, we add KL divergence term with the balance factor $\gamma$. It is worth noting that the gap between $\mathcal{L}_{IBC}$ and $\mathcal{L}_{UIB}$ lies in the preservation of the clustering structure in the process of information compression.

**Connection to DEC**. Deep Embedding for Clustering (DEC) [12] begins with pre-training an auto-encoder to learn low-dimensional representations, and the soft cluster assignment computed by the embeddings (as $q(\boldsymbol{c}|\mathbf{z})$ in Eq. (3)) is determined by the Student's $t$-distribution. The clustering loss is designed via KL divergence to approach the auxiliary target distribution (as $p(\boldsymbol{c}|\mathbf{x})$ in Eq. (3)). By setting $\beta = 0$ and $\gamma = 1$, and using the pre-training strategy, we obtain the DEC model.

**Connection to VaDE**. Variational Deep Embedding (VaDE) [13] combines VAEs and Gaussian mixture models and encodes the initial data distribution as a learned Gaussian mixture distribution in the latent variable space. By setting $\beta = 1$ and $\gamma = 1$, we get the VaDE model.

## 3 Information-based Multi-view Clustering

Let $\mathcal{X} = \{\mathbf{X}^{(v)}\}_{v=1}^{m}$ be a dataset consisting of $m$ views, where each view collection comprises $n$ independent and identically distributed (i.i.d) samples, $\mathbf{X}^{(v)} = \{\mathbf{x}_1^{(1)}, \mathbf{x}_2^{(2)}, ..., \mathbf{x}_n^{(v)}\}$. Consider $\mathcal{Z} = \{\mathbf{Z}^{(v)}\}_{v=1}^{m}$ presenting view-peculiar compressed representations for $\mathcal{X}$ and $\boldsymbol{c} \in \{1, ..., K\}$ designating a categorical random variable and represent the index of the actual cluster. For multi-view clustering, we first learn a unified latent representation $\mathbf{Z}$ across all views, which satisfies comprehensiveness, concentration and cross-diversity. Additionally, the multi-view representation is needed to preserve the original data structure information related to clusters followed by performing clustering.

### 3.1 Unified multi-view representation learning

We first define the desiderata for multi-view representation as follows.

**Definition 3.1** (**Comprehensive, concentrative and cross-diverse multi-view representation**).
**Comprehensiveness**. Given the multi-view observations $\mathcal{X}$, a representation, $\mathbf{Z}$, is comprehensive if each view observation can be predicted by $\mathbf{Z}$,

$$\mathbf{Z}^* = \arg\max_{\mathbf{Z}} I(\mathbf{Z}; \mathcal{X}). \qquad (4)$$

**Concentration**. For each view data $\mathbf{X}^{(v)}$, their view-peculiar compressed representation $\mathbf{Z}^{(v)}$ is concentrative if the redundant information for each view is eliminated,

$$\mathbf{Z}^{(v)*} = \arg\min_{\mathbf{Z}^{(v)}} I(\mathbf{Z}^{(v)}; \mathbf{X}^{(v)}). \qquad (5)$$

**Cross-diversity**. View-peculiar latent representations $\mathcal{Z}$ can be cross-diverse if they avoid duplication of information within the view,

$$\mathcal{Z}^* = \arg\max_{\mathcal{Z}} I(\mathbf{Z}; \mathcal{Z}). \qquad (6)$$

To more intuitively grasp the definition of multi-view representations, we depict the three requirements for representation learning from the perspective of information theory. As illustrated in Fig 1, taking dual-view data as an example, the information carried by $\mathbf{X}^{(1)}$ and $\mathbf{X}^{(2)}$ are $H(\mathbf{X}^{(1)})$ and $H(\mathbf{X}^{(2)})$ respectively, which are fixed. To effectively convey the combination of the two views, the unified representation $\mathbf{Z}$ achieves informativeness by maximizing the mutual information between $\mathbf{Z}$ and

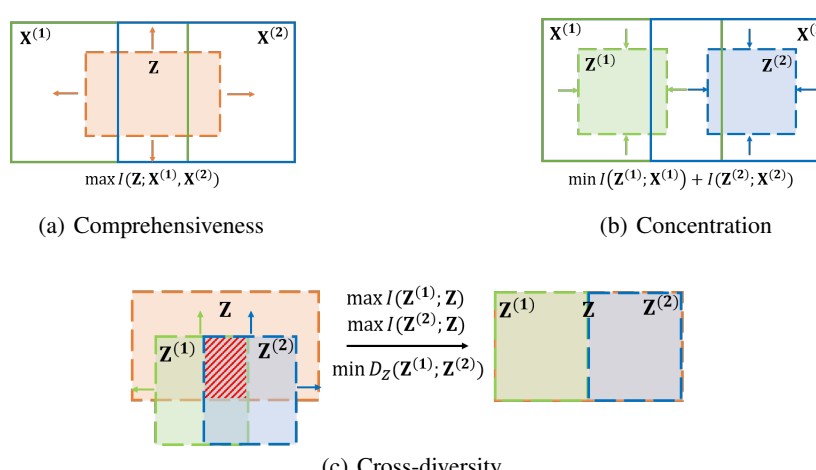

(a) Comprehensiveness

(b) Concentration

(c) Cross-diversity

Figure 1: Illustration of three desiderata for learning multi-view representations.

$\mathbf{X}^{(1)}\mathbf{X}^{(2)}$. Additionally, to remove redundant information from each view, we minimize $I(\mathbf{Z}^{(1)}; \mathbf{X}^{(1)})$ and $I(\mathbf{Z}^{(2)}; \mathbf{X}^{(2)})$ respectively for maximum compression of the respective view information. Ideally, semantically irrelevant information specific to each view is removed while complete semantically relevant information is retained.

To balance comprehensiveness and concentration, we introduce mutual information between $\mathbf{Z}$ and $\mathbf{Z}^{(1)}\mathbf{Z}^{(2)}$ as a cross-diversity indicator,

$$I(\mathbf{Z}; \mathbf{Z}^{(1)}, \mathbf{Z}^{(2)}) = I(\mathbf{Z}; \mathbf{Z}^{(1)}) + I(\mathbf{Z}; \mathbf{Z}^{(2)}) - \left[ I(\mathbf{Z}^{(1)}; \mathbf{Z}^{(2)}) - I(\mathbf{Z}^{(1)}; \mathbf{Z}^{(2)}|\mathbf{Z}) \right], \qquad (7)$$

where we denote $D_{\mathbf{Z}}(\mathbf{Z}^{(1)}; \mathbf{Z}^{(2)}) = I(\mathbf{Z}^{(1)}; \mathbf{Z}^{(2)}) - I(\mathbf{Z}^{(1)}; \mathbf{Z}^{(2)}|\mathbf{Z})$, representing the duplicate information between views. As showed in Fig 1 (c), one can approach cross-diversity by maximizing $I(\mathbf{Z}; \mathbf{Z}^{(1)})$, $I(\mathbf{Z}; \mathbf{Z}^{(2)})$ and minimizing $D_{\mathbf{Z}}(\mathbf{Z}^{(1)}; \mathbf{Z}^{(2)})$.

**Proposition 3.1 (Cross-diversity is sufficient for comprehensiveness and concentration, respectively).** *Given two Markov chains $\mathbf{X}^{(1)} \multimap \mathbf{Z}^{(1)} \multimap \mathbf{Z}$ and $\mathbf{X}^{(2)} \multimap \mathbf{Z}^{(2)} \multimap \mathbf{Z}$, maximization of $I(\mathbf{Z}; \mathbf{Z}^{(1)}, \mathbf{Z}^{(2)})$ is sufficient for maximization of $I(\mathbf{Z}; \mathbf{X}^{(1)}, \mathbf{X}^{(2)})$ when $\mathbf{Z}^{(1)}, \mathbf{Z}^{(2)}$ are fixed and minimization of $\sum_{v=1}^{2} I(\mathbf{Z}^{(v)}; \mathbf{X}^{(v)})$ when $\mathbf{Z}$ is fixed, respectively.*

*Proof.* Under the two Markov chains accompanied by loss of information, we have,

$$I(\mathbf{Z}; \mathbf{X}^{(1)}, \mathbf{X}^{(2)}) \overset{(b)}{\leq} I(\mathbf{Z}^{(1)}; \mathbf{X}^{(1)}) + I(\mathbf{Z}^{(2)}; \mathbf{X}^{(2)}),$$

Next we rewrite the cross-diversity indicator,

$$\begin{aligned} I(\mathbf{Z}; \mathbf{Z}^{(1)}, \mathbf{Z}^{(2)}) &= H(\mathbf{Z}^{(1)}, \mathbf{Z}^{(2)}) - H(\mathbf{Z}^{(1)}, \mathbf{Z}^{(2)}|\mathbf{Z}) \\ &= H(\mathbf{Z}^{(1)}) + H(\mathbf{Z}^{(2)}) - I(\mathbf{Z}^{(1)}; \mathbf{Z}^{(2)}) - H(\mathbf{Z}^{(1)}, \mathbf{Z}^{(2)}|\mathbf{Z}) \\ &\leq H(\mathbf{Z}^{(1)}) + H(\mathbf{Z}^{(2)}), \end{aligned}$$

Since $I(\mathbf{Z}; \mathbf{Z}^{(1)}, \mathbf{Z}^{(2)})$ achieves the maximum value, i.e. $I(\mathbf{Z}; \mathbf{Z}^{(1)}, \mathbf{Z}^{(2)}) = H(\mathbf{Z}^{(1)}) + H(\mathbf{Z}^{(2)})$. We denote $I(\mathbf{Z}; \mathbf{Z}^{(1)}, \mathbf{Z}^{(2)})$ as $\mathbf{T}$,

$$\begin{aligned} I(\mathbf{Z}; \mathbf{X}^{(1)}, \mathbf{X}^{(2)}) &= I(\mathbf{T}; \mathbf{X}^{(1)}, \mathbf{X}^{(2)}) \\ &= H(\mathbf{T}) - H(\mathbf{T}|\mathbf{X}^{(1)}, \mathbf{X}^{(2)}) \\ &\overset{(c)}{\geq} H(\mathbf{Z}^{(1)}) + H(\mathbf{Z}^{(2)}) - H(\mathbf{Z}^{(1)}, \mathbf{Z}^{(2)}|\mathbf{X}^{(1)}, \mathbf{X}^{(2)}) \\ &= H(\mathbf{Z}^{(1)}) - H(\mathbf{Z}^{(1)}|\mathbf{X}^{(1)}) + H(\mathbf{Z}^{(2)}) - H(\mathbf{Z}^{(2)}|\mathbf{X}^{(2)}) \\ &= I(\mathbf{Z}^{(1)}; \mathbf{X}^{(1)}) + I(\mathbf{Z}^{(2)}; \mathbf{X}^{(2)}), \end{aligned}$$

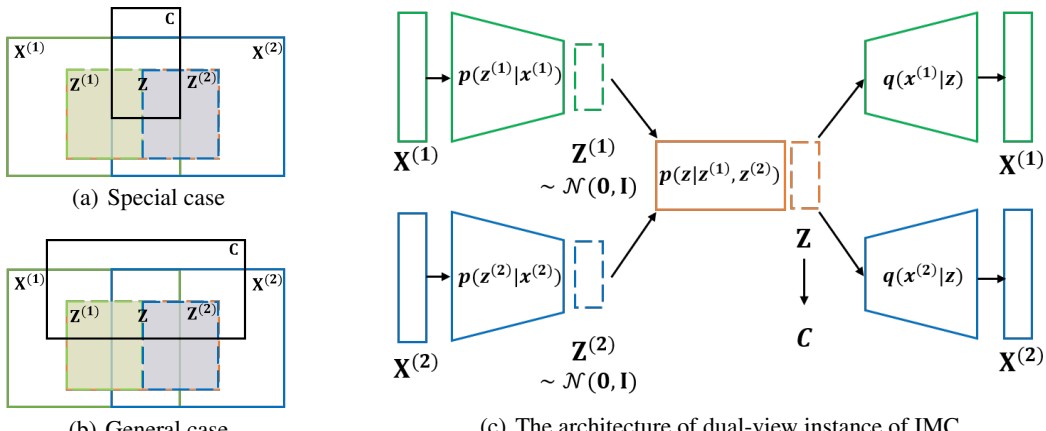

(a) Special case

(b) General case

(c) The architecture of dual-view instance of IMC

Figure 2: (a) Cluster information is only related to common information between multiple views. (b) View-peculiar information also includes semantic features. (c) Multi-view variational autoencoders and a fusion network are incorporated to perform representation learning and clustering simultaneously.

To satisfy both inequalities $(b)$ and $(c)$, the equation holds, i.e., $I(\mathbf{Z}; \mathbf{X}^{(1)}, \mathbf{X}^{(2)}) = I(\mathbf{Z}^{(1)}; \mathbf{X}^{(1)}) + I(\mathbf{Z}^{(2)}; \mathbf{X}^{(2)})$. Therefore, when $I(\mathbf{Z}; \mathbf{Z}^{(1)}, \mathbf{Z}^{(2)})$ is maximized, fixing $\mathbf{Z}^{(1)}, \mathbf{Z}^{(2)}, I(\mathbf{Z}; \mathbf{X}^{(1)}, \mathbf{X}^{(2)})$ maximization holds, and conversely, fixing $\mathbf{Z}$, $I(\mathbf{Z}^{(1)}; \mathbf{X}^{(1)}) + I(\mathbf{Z}^{(2)}; \mathbf{X}^{(2)})$ minimization holds.  $\square$

With these desiderata, the unified multi-view representation learning can be formulated as follows,

$$\max_{\mathbf{Z},\boldsymbol{\mathcal{Z}}} I(\mathbf{Z}; \boldsymbol{\mathcal{X}}) - \sum_{v}^{m} I(\mathbf{Z}^{(v)}; \mathbf{X}^{(v)}) + \beta I(\mathbf{Z}; \boldsymbol{\mathcal{Z}})$$

$$\overset{(d)}{\geq} \mathbb{E}_{p(\boldsymbol{\mathcal{X}})} \left[ \mathbb{E}_{p(\mathbf{z}|\boldsymbol{\mathcal{X}})} [\log q(\boldsymbol{\mathcal{X}}|\mathbf{z})] - \sum_{v}^{m} D_{KL}(p(\mathbf{z}^{(v)}|\mathbf{x}^{(v)})||q(\mathbf{z}^{(v)})) \right] + \beta I(\mathbf{Z}; \boldsymbol{\mathcal{Z}}), \quad (8)$$

where $(d)$ leverages the variational technique to construct the lower bound of the first two terms (deviation can be found in Appendix B), the $\beta$ is the balance coefficient that controls the cross-diversity of representation and balances the comprehensiveness and concentration. Mutual information between $\mathbf{Z}$ and multiple representations $\mathbf{Z}^{(1)}, ..., \mathbf{Z}^{(m)}$ can further be expressed by,

$$I(\mathbf{Z}; \boldsymbol{\mathcal{Z}}) = \sum_{v=1}^{m} I(\mathbf{Z}; \mathbf{Z}^{(v)}) - \sum_{v=1}^{m} \sum_{t \neq v}^{m} D_{\mathbf{Z}}(\mathbf{Z}^{(v)}, \mathbf{Z}^{(t)}). \quad (9)$$

### 3.2 Generalized multi-view clustering framework

For the multi-view clustering task, previous works [14, 15] considered category-related or semantic-related information to be relevant only to the multi-view consensus part. As shown in Fig. 2 (a), the part where two views intersect contains all the information related to cluster $c$ in the multi-view data. However, this is only a special case. More generally, the respective views contain specific semantic information that can complement each other. This is why a comprehensive understanding of an object or event requires different views. As shown in Fig. 2 (b), we cannot determine if the view-common or view-peculiar parts of the views contain more semantic information, we can only assume that they all contain this information.

Next, we seek to learn the unified multi-view representation without losing the original clustering structure of the data. For examples, we assume that the data obeys a Gaussian mixture distribution, or that the data has a tight intra-class structure and a relaxed inter-class structure. We add a clustering term to Equation (8) for measuring the distance between the posterior distribution of a given original

data to clusters and the posterior distribution of a given multi-view representation to clusters. Then, the objective of information-based multi-view clustering can be formulated as,

$$\mathcal{L}_{IMC} = \mathbb{E}_{p(\mathcal{X})} \Big[ \underbrace{\mathbb{E}_{p(\mathbf{z}|\mathcal{X})}[\log q(\mathcal{X}|\mathbf{z})]}_{\text{data reconstruction}} - \underbrace{\sum_{v}^{m} D_{KL}(p(\mathbf{z}^{(v)}|\mathbf{x}^{(v)})||q(\mathbf{z}^{(v)}))}_{\text{multi-regularization}}$$

$$- \gamma \underbrace{\mathbb{E}_{p(\mathbf{z}|\mathcal{X})}[D_{KL}(p(\mathbf{c}|\mathcal{X})||q(\mathbf{c}|\mathbf{z}))]}_{\text{clustering}} \Big] + \beta \underbrace{I(\mathbf{Z};\mathcal{Z})}_{\text{information shift}} . \tag{10}$$

Note that Equation (10) can be regarded as four items representing specific roles: data reconstruction, multi-regularization, information shift and clustering. The data reconstruction item maximizes the expectation of the log-likelihood to achieve comprehensive so that the multi-view representation retain more information of the data. The multi-regularization term makes the multi-view representation forget the information of the original data by minimizing the KL divergence of the posterior distribution and the variational distribution to meet the requirement of concentration. Information shift item adjusts the information relationship between multi-view representations to achieve cross-diverse. The clustering item measures the degree to which the data retains the original data structure after dimensionality reduction.

### 3.3 Numerical scheme to solve IMC

Following VAE [16], we use a deep neural network to estimate the parameters of the unknown distribution to instantiate IMC, as shown in Fig. 2 (c). According to the Markov chain $\mathbf{X}^{(v)} \multimap \mathbf{Z}^{(v)} \multimap \mathbf{Z} \multimap \mathbf{X}^{(v)}$, the posterior $p(\mathbf{z}|\mathcal{X})$ and likelihood $q(\mathcal{X}|\mathbf{z})$ can be factorized as, respectively,

$$p(\mathbf{z}|\mathcal{X}) = \int \int ... \int p_\psi(\mathbf{z}|\{\mathbf{z}^{(v)}\}_{v=1}^m) \prod_{v=1}^{m} p_{\theta^{(v)}}(\mathbf{z}^{(v)}|\mathbf{x}^{(v)}) d_{\mathbf{z}^{(1)}} d_{\mathbf{z}^2}...d_{\mathbf{z}^m},$$

$$= \mathbb{E}_{p_{\theta^{(1)}}(\mathbf{z}^{(1)}|\mathbf{x}^{(1)})} \mathbb{E}_{p_{\theta^{(2)}}(\mathbf{z}^{(2)}|\mathbf{x}^{(2)})} ... \mathbb{E}_{p_{\theta^{(m)}}(\mathbf{z}^{(m)}|\mathbf{x}^{(m)})} \Big[ p_\psi(\mathbf{z}|\{\mathbf{z}^{(v)}\}_{v=1}^m) \Big],$$

$$q(\mathcal{X}|\mathbf{z}) = \prod_{v=1}^{m} q_{\phi^{(v)}}(\mathbf{x}^{(v)}|\mathbf{z}),$$

where $p_{\theta^{(v)}}(\mathbf{z}^{(v)}|\mathbf{x}^{(v)})$ denotes the view-peculiar encoder with parameter $\theta^{(v)}$, $q_{\phi^{(v)}}(\mathbf{x}^{(v)}|\mathbf{z}^{(v)})$ denotes the view-peculiar decoder with parameter $\phi^{(v)}$. To obtain the unified multi-view representations from view-peculiar representations, we design a fusion module $p_\psi(\mathbf{z}|\{\mathbf{z}^{(v)}\}_{v=1}^m)$ parameter with multiple fully connected layers as $\psi$.

For clustering item, we adopt the same strategy as DEC [12], replace $q(\mathbf{c}|\mathbf{z})$ with Student's t-distribution, and set target $p(\mathbf{c}|\mathcal{X})$ with auxiliary distribution,

$$q_{ik} = \frac{(1 + \|\mathbf{z}_i - \mathbf{u}_k\|^2)^{-1}}{\sum_j (1 + \|\mathbf{z}_i - \mathbf{u}_j\|^2)^{-1}},$$

$$p_{ik} = \frac{q_{ik}^2 / \sum_i q_{ik}}{\sum_j (q_{ij}^2 / \sum_i q_{ij})}, \tag{11}$$

where $q_{ik}$ measures the similarity between the representation $\mathbf{z}_i$ and cluster center $\mathbf{u}_k$. $p_{ik}$ enhances the data points assigned with high confidence to improve the purity of the cluster.

For optimization IMC using Stochastic Gradient Variational Bayes (SGVB), the sampling operations of $(\{\mathbf{z}^{(v)}\}_{v=1}^m, \mathbf{z})$ should be mapped to the deterministic functions. According to the *reparameterization* trick [16] for the continuous variable, we sample the $t$-th latent representations by,

$$\mathbf{z}_t^{(v)} = \boldsymbol{\mu}_{\phi^{(v)}} + \boldsymbol{R}_{\phi^{(v)}} \boldsymbol{\epsilon}_t^{(v)}, \quad \text{where } \boldsymbol{R}_{\phi^{(v)}} \boldsymbol{R}_{\phi^{(v)}}^T = \boldsymbol{\Sigma}_{\boldsymbol{\phi}^{(v)}}, \; \boldsymbol{\epsilon}_t^{(v)} \in \mathcal{N}(\mathbf{0}, \mathbf{I}) \tag{12}$$

$$\mathbf{z}_t = \boldsymbol{\mu}_\psi + \boldsymbol{R}_\psi \boldsymbol{\epsilon}_t, \qquad \text{where } \boldsymbol{R}_\psi \boldsymbol{R}_\psi^T = \boldsymbol{\Sigma}_\psi, \; \boldsymbol{\epsilon}_t \in \mathcal{N}(\mathbf{0}, \mathbf{I}) \tag{13}$$

---
**Algorithm 1** Optimization Procedure of IMC
---
**Input**: Multi-view dataset $\mathcal{X}$; Setting $T = 1$ and dimensionality of the latent variables.
**Parameter**: Initialize parameters $\{\phi^{(v)}\}$, $\{\theta^{(v)}\}$, $\psi$ with random values, cluster centers $\mathbf{u} = \{\mathbf{u}_1, ..., \mathbf{u}_K\}$ by $K$-means.
 1: **while** not reaching the maximal epochs **do**
 2:     **for** $i$ in $n$ samples **do**
 3:         **for** $v$ in $m$ views **do**
 4:             Calculate $(\boldsymbol{\mu}_{\phi^{(v)}}, \boldsymbol{\Sigma}_{\phi^{(v)}})$ through $v$-th encoder and then sample $\mathbf{z}_{it}^{(v)}$ by Equation (12);
 5:         **end for**
             Calculate $(\boldsymbol{\mu}_\psi, \boldsymbol{\Sigma}_\psi)$ through the fusion module and then sample $\mathbf{z}_{it}$ by Equation (13);
 6:         **for** $v$ in $m$ views **do**
 7:             Generate $\mathbf{x}^{(v)}$ by $v$-th decoders.
 8:         **end for**
 9:     **end for**
10:     Update $\{\phi^{(v)}\}$, $\{\theta^{(v)}\}$, $\psi$, $\mathbf{u}$ by maximizing Equation (14).
11: **end while**
**Output**: The cluster for each sample $\boldsymbol{c}_i$ by selecting the index of maximum probability value $q(\boldsymbol{c}_i | \mathbf{z}_i)$.
---

where $\boldsymbol{\mu}_{\phi^{(v)}}$ and $\boldsymbol{R}_{\phi^{(v)}}$ are the outputs of the $v$-th encoder, $\boldsymbol{\mu}_\psi$ and $\boldsymbol{R}_\psi$ are the outputs of the fusion module, respectively.

By using Monte-Carlo estimators, the objective of IMC can be further written as,

$$
\max_{\{\theta^{(v)}\},\{\phi^{(v)}\},\psi,\mathbf{u}} \mathcal{L}_{\text{IMC}} = \frac{1}{n} \sum_{i=1}^{n} \left[ \frac{1}{T} \sum_{v=1}^{m} \sum_{t=1}^{T} \left[ \log q_{\phi^{(v)}}(\mathbf{x}_i^{(v)} | \mathbf{z}_{it}) - \log p_{\theta^{(v)}}(\mathbf{z}_{it}^{(v)} | \mathbf{x}_i^{(v)}) \right. \right.
$$

$$
\left. \left. + \log \mathcal{N}(\mathbf{0}, \mathbf{I}) \right] - \gamma \sum_{k=1}^{K} p_{ik} \log \frac{p_{ik}}{q_{ik}} \right] + \beta I_{JS}(\mathbf{Z}; \boldsymbol{\mathcal{Z}}), \tag{14}
$$

where $T$ denotes the number of Monte Carlo samples and is usually set to be 1. Additionally, variational distribution $q(\mathbf{z}^{(v)})$ is set to multivariate standard normal distribution, i.e., $q(\mathbf{z}^{(v)}) = \mathcal{N}(\mathbf{0}, \mathbf{I})$. The mutual information between representations $I(\mathbf{z}; \boldsymbol{\mathcal{Z}})$ can be maximized by using sample-based differentiable mutual information lower bound, Jensen-Shannon estimator [17]. The partial derivatives of each parameter combination are then calculated for the stochastic backpropagation technique. The concrete optimization procedure of IMC is summarized in Algorithm 1.

# 4 Related Work

Recently, several works have been devoted to the problem of multi-view representation learning based on information theory. For supervised methods, [18] analyses the consistency and complementary characteristics of multi-view features from the information bottleneck theory, and strengthen the discriminative power of the encoder through the margin maximization method to balance the accuracy and complexity of the multi-view model. [19] seeks to fuse the features of each view to obtain a joint representation through a deep neural network, and maximize the mutual information between labels and joint representations to preserve the target information. Given a graph and its labels, [20] proposes a Graph Information Bottleneck (GIB) framework to efficiently infer the most informative yet compressed subgraph for recognition.

While in the unsupervised scheme, [14] proposes a multi-view representation learning method based on information bottleneck theory, Multi-view Information Bottleneck (MIB). MIB assumes that the shared part of the two views is the target information, and the part not shared by the two views is the superfluous information, followed by leveraging off-the-shelf information bottleneck technique to learn robust representations. [21] explores the potential relationship and intrinsic information between different views, and discarded redundant information from multi-view data with the help of the information bottleneck principle to balance the consistency and complementarity between multiple views. Under the framework of information theory, Completer [22] maximizes multi-view shared mutual information for consistent learning, and minimizes conditional entropy to recover missing view with available data. [23] regards each view source data as a self-supervised signal for

learning latent representations, removing redundant information from the view itself while retaining consistent information for shared features across multiple views. A multivariate mutual information term is employed to decompose view-specific representations, which further ensures that the multi-view fusion self-expression matrix via view-common features is compact and informative. The aforementioned unsupervised multi-view representation learning methods aim to learn consistent information shared across multiple views, as shown in Fig. 2 (a). These methods either have limitations in practical applications or do not consider the duplicate information among views. It motivates us to construct a general information-theoretic-based unsupervised multi-view clustering framework and introduce three requirements for multi-view representation learning.

## 5 Experiments

### 5.1 Experimental settings

**Model setup**: The architectures of $p_{\phi^{(v)}}(\mathbf{z}^{(v)}|\mathbf{x}^{(v)})$ and $q_{\theta^{(v)}}(\mathbf{x}^{(v)}|\mathbf{z})$ are fully connected networks with with $d_v$-500-500-1204-256 and 256-1024-500-500-$d_v$ neurons, respectively, where $d_v$ is the dimensionality of each view. The fusion network $p_{(\psi)}(\mathbf{z}|\{\mathbf{z}^{(v)}\}_{v=1}^m)$ concatenates multiple view-peculiar latent representations, followed by a fully connected layer. Adam optimizer [24] is utilized to maximize the objective function, and set the learning rate to be 0.001 with a decay of 0.9 for every 10 epochs.

**Datasets**: We adopt four real-world datasets listed as follows, (1) **UCI-digits** [25] contains 2000 examples of ten numerals from 0 to 9 with five views which are respectively extracted by Fourier coefficients, profile correlations, Karhunen-Love coefficient, Zernike moments, and pixel average extractors. (2) **Notting-Hill** [26] is widely used video face dataset for clustering, which collects 4660 faces across 76 tracks of the 5 main actors from the movie 'Notting Hill'. We use the multi-view version provided in [27], consisting of 550 images with three kind of features, i.e., LBP, gray pixels, and Gabor features. (3) **BDGP** [28] contains 2500 images in 5 categories, and each sample is described by a 1750-D image vector and a 79-D textual feature vector. (4) **Caltech20** is a subset of the object recognition dataset [35] containing 20 classes with six different views, including Gabor features, wavelet moments, CENTRIST features, histogram of oriented gradients, GIST features and local binary patterns.

**Compared algorithms**: Four state-of-the-art algorithms and two ablation models are used to compare the clustering performance, including: (1) **DMVAE** [36] assumes that the data and multi-view latent representation obey a Gaussian mixture distribution, which can be considered as a special case of IMC, i.e., let $\beta = 0$ and $\gamma = 1$. (2) **MIB** [14] considers the parts shared by both views as target information and the view-specific parts as superfluous information. (3) **CMIB-Nets** [21] proposes collaborative multi-view information bottlenecks to learn the information inherent within a view and the shared structure between views, while reducing redundant information. (4) **Completer** [15] introduces maximization of mutual information between views and minimization of conditional entropy of different views to achieve dual prediction and representation learning. For fairness, we repeatedly perform all compared algorithms 10 times on four datasets and the mean and standard deviation were recorded.

**Ablation models**: For ablation study, we construct two variants of IMC as a comparison. (1) **IMC-v1** indicates that the objective function Equation (10) reduces the item of information shift, i.e., $\beta = 0$. (2) **IMC-v2** sets $\gamma = 0$ to discard the KL divergence of the clustering item, and uses $k$-means to perform clustering on the unified multi-view representation.

**Evaluation metrics**: For a comprehensive analysis, we use three popular clustering metrics including Normalized Mutual Information (NMI), Accuracy (ACC) and adjusted rand index (ARI). The higher the values of these indicators, the better the clustering performance.

### 5.2 Performance evaluation

We tested seven methods on four multi-view datasets. The experimental results are summarized in Table 1. It can be observed that i) combining the four datasets in terms of three indicators, IMC has obtained the best performance 8 times and the second-best performance 3 times, which is the most stable and accurate multi-view clustering model. ii) Although the Completer model has achieved the

Table 1: Clustering performance comparison on four datasets (mean±standard deviation). The optimal and suboptimal results are in bold and underlined, respectively.

| Datasets | Metrics | DMVAE | MIB | CMIB-Nets | Completer | IMC-v1 | IMC-v2 | IMC |
|---|---|---|---|---|---|---|---|---|
| UCI-digits | ACC | 90.95±0.62 | 83.30±1.27 | 85.70±1.15 | 91.28±1.41 | 90.01±0.60 | 84.01±1.15 | **92.13±0.55** |
| | NMI | 85.54±1.06 | 75.43±1.04 | 78.31±1.41 | 86.34±0.60 | 84.79±0.32 | 79.01±1.54 | **88.01±0.73** |
| | ARI | 85.40±1.54 | 76.16±1.55 | 76.97±1.64 | 86.67±0.86 | 84.78±0.43 | 78.18±0.88 | **87.83±0.25** |
| Notting-Hill | ACC | 76.22±1.21 | 81.65±1.37 | 85.40±2.36 | 80.17±2.79 | 77.79±2.00 | 84.83±1.90 | **87.10±1.35** |
| | NMI | 72.97±0.96 | 75.95±2.52 | 78.65±2.57 | 76.11±2.27 | 74.93±1.94 | 78.79±1.08 | **80.67±1.42** |
| | ARI | 69.50±0.77 | 71.91±1.61 | **80.46±1.92** | 71.48±3.29 | 70.30±2.15 | 79.10±2.32 | 80.19±1.74 |
| BDGP | ACC | 90.59±1.45 | 86.82±0.65 | 85.82±0.58 | 79.31±1.55 | 88.70±1.42 | 80.46±1.85 | **91.46±0.82** |
| | NMI | **85.32±0.53** | 80.82±0.86 | 81.60±0.66 | 74.25±0.69 | 81.47±1.12 | 73.31±2.45 | 84.40±1.20 |
| | ARI | 78.58±2.54 | 73.90±1.65 | 74.25±2.19 | 71.44±1.45 | 77.65±2.15 | 69.31±2.80 | **80.25±1.62** |
| Caltech20 | ACC | 61.50±0.54 | 56.12±2.54 | 55.26±3.18 | **62.31±2.65** | 58.05±1.28 | 52.42±3.15 | 60.82±1.66 |
| | NMI | 68.32±1.23 | 63.28±2.66 | 62.44±2.56 | **70.25±2.20** | 65.75±2.20 | 61.76±3.40 | 69.20±1.48 |
| | ARI | 59.86±1.46 | 58.10±2.90 | 56.45±2.74 | 61.14±2.55 | 57.52±2.56 | 53.30±2.82 | **61.42±2.12** |

Table 2: Comparison of NMI at noise ratios of 10%, 30%, and 50% (mean±standard deviation). The optimal and suboptimal results are in bold and underlined, respectively.

| Datasets | Noise | DMVAE | MIB | CMIB-Nets | Completer | IMC-v1 | IMC-v2 | IMC |
|---|---|---|---|---|---|---|---|---|
| UCI-digits | 10% | 82.35±1.24 | 74.60±1.75 | 75.55±2.35 | 85.28±1.84 | 81.12±2.26 | 78.46±1.45 | **86.82±1.05** |
| | 30% | 75.84±2.26 | 66.35±3.45 | 65.50±2.82 | 73.40±2.56 | 69.90±3.25 | 72.18±2.48 | **78.56±1.36** |
| | 50% | 63.80±1.58 | 58.46±2.80 | 56.46±1.64 | 60.76±3.65 | 56.84±3.40 | 65.30±2.66 | **70.35±2.28** |

best NMI performance in caltech20, its NMI performance lags behind by 12.97% in BDGP. This instability can also be found in the MIB model. iii) The end-to-end model (IMC, IMC-v1, DMVAE) that considers the clustering loss in the process of learning multi-view representation will achieve better clustering results than the two-stage model (MIB, CMIB-nets, Completer). iv) Comparing IMC, IMC-v1 and IMC-v2, the performance of using KL divergence clustering term has been significantly improved, while the use of information shift term has a small improvement.

We therefore draw three conclusions: First, the IMC developed based on the general definition of multi-view information theory obtains more stable clustering results. Second, end-to-end models help to learn more discriminative representations. Third, in the process of learning multi-view representations, three requirements (comprehensiveness, concentration, cross-diversity) play a role in improving the clustering performance.

### 5.3 Robustness Assessment

To evaluate the robustness of the model, we add different scales of sparse noise to the dataset UCI-digits. Specifically, we randomly select 10%, 30%, and 50% pixels of the original image size, and add pepper noise or salt noise to each pixel with probability 0.5, i.e., set 0 or 255.

As shown by the clustering results in Table 2, the performance of all algorithms decreases as the noise ratio increases, but the performance of our algorithm is always the best. It can be observed that the decline of IMC is the smallest, while the decline of the Completer model is obvious. This may be due to the assumption of multi-view semantic consistency is more sensitive as the noise ratio increases, the information shared by multiple views will also incorporate more noise. It is worth noting that the performance of IMC-v2 is better than that of IMC-v1, which is contrary to the clustering experiment in Table 1. It can be concluded that only pursuing inter-view consistent information leads to semantic bias when the dataset contains noise, while more informative representations can be obtained for robustness based on the general multi-view information bottleneck theory.

### 5.4 Parameter analysis

The proposed method contains two balance parameters, the information shift balance coefficient $\beta$, and the clustering loss trade-off parameter $\gamma$. Although IMC at fixed values of some parameters has

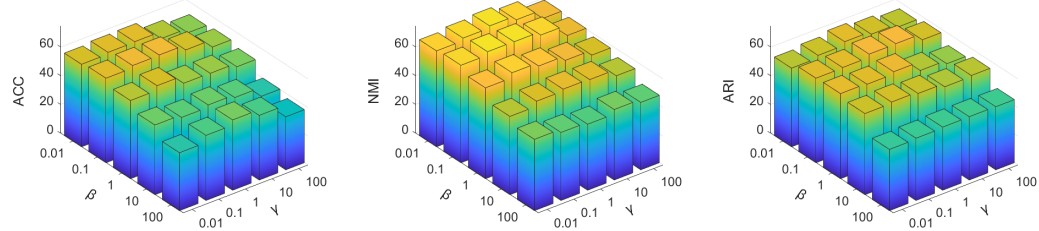

Figure 3: Parameter analysis of $\beta$ and $\gamma$ on Caltech20 dataset.

shown satisfactory performance, it is still important to explore the effect of varying these parameters on the performance and to mine the inner mechanism of this model. We evaluate the parameter sensitivity of ACC, NMI, ARI metrics on Caltech20 dataset, as shown in Figure 3, we choose the values of $\beta$ and $\gamma$ in the range of $\{0.01, 0.1, 1, 10, 100\}$. It can be seen from the results that our method is not sensitive to the coefficient of clustering loss, but how to adjust the parameter $\beta$ is the key to improving clustering performance. As expressed in Equation (8), $\beta$ controls the degree of cross-diversity of multi-view representation. Therefore, the selection strategy of the beta parameter can be based on prior knowledge of the dataset. In our experience, if there is a large degree of consistency in the semantics of multiple views, then the $\beta$ is set to small and vice versa.

## 6 Conclusion

In this paper, we propose an information-theoretic multi-view clustering framework that avoids the assumption of semantic consistency across multiple views. Specifically, we extend the information bottleneck theory to unsupervised multi-view learning through defining three requirements for multi-view representations via mutual information. By constructing a multi-view variational lower bound and introducing KL divergence as a clustering loss, the entire framework is finally optimized by deep neural network and stochastic gradient variational Bayes. Experiments verify the superiority and robustness of the generalized information-based multi-view clustering on four benchmark datasets and noisy data.

The practical limitations of the proposed model lie in the choice of parameters. The mathematical strategy for selecting the optimal parameters is a direction that can be studied in the future. In addition, constructing a tighter lower bound or estimator to approximate high-dimensional mutual information is also a place for further improvement in this work.

## Acknowledgment

We thank the anonymous reviewers for their helpful feedbacks. This work was supported in part by the National Key Research and Development Program of China (2022YFE0112200), the Key-Area Research and Development of Guangdong Province (2022A0505050014, 2022B1111050002), the Key-Area Research and Development Program of Guangzhou City (202206030009, 2023B01J0002), the National Natural Science Foundation of China (U21A20520, 62172112), Guangdong Key Laboratory of Human Digital Twin Technology (2022B1212010004).

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

## Appendix A. Derivation of Equation (2)

By substituting the intractable posterior $p(\mathbf{x}|\mathbf{z})$ and marginal $p(\mathbf{z})$ with respective variational posterior $q(\mathbf{x}|\mathbf{z})$ and marginal $q(\mathbf{z})$, we form a variational lower bound for unsupervised information bottleneck,

$$
\max_{\mathbf{Z}} I(\mathbf{Z}; \mathbf{X}) - \beta I(\mathbf{Z}; i)
$$

$$
= \int \int p(\mathbf{z}, \mathbf{x}) \log \frac{p(\mathbf{x}|\mathbf{z})}{p(\mathbf{x})} d\mathbf{x} d\mathbf{z} - \beta \sum_i \int p(\mathbf{z}, i) \log \frac{p(\mathbf{z}|i)}{p(\mathbf{z})} d\mathbf{z}
$$

$$
= H(\mathbf{x}) + \mathbb{E}_{p(\mathbf{x})} \left[ \int p(\mathbf{z}|\mathbf{x}) \log p(\mathbf{x}|\mathbf{z}) d\mathbf{z} \right] - \beta \frac{1}{n} \sum_i \int p(\mathbf{z}|\mathbf{x}_i) \log \frac{p(\mathbf{z}|\mathbf{x}_i)}{p(\mathbf{z})} d\mathbf{z}
$$

$$
\geq \mathbb{E}_{p(\mathbf{x})} \left[ \int p(\mathbf{z}|\mathbf{x}) \log q(\mathbf{x}|\mathbf{z}) d\mathbf{z} \right] - \beta \frac{1}{n} \sum_i \int p(\mathbf{z}|\mathbf{x}_i) \log \frac{p(\mathbf{z}|\mathbf{x}_i)}{q(\mathbf{z})} d\mathbf{z}
$$

$$
= \mathbb{E}_{p(\mathbf{x})} \left[ \int p(\mathbf{z}|\mathbf{x}) \log q(\mathbf{x}|\mathbf{z}) d\mathbf{z} - \beta \int p(\mathbf{z}|\mathbf{x}) \log \frac{p(\mathbf{z}|\mathbf{x})}{q(\mathbf{z})} d\mathbf{z} \right]
$$

$$
= \mathbb{E}_{p(\mathbf{x})} \left[ \mathbb{E}_{p(\mathbf{z}|\mathbf{x})} \left[ \log q(\mathbf{x}|\mathbf{z}) \right] - \beta D_{KL} \left( p(\mathbf{z}|\mathbf{x}) \| q(\mathbf{z}) \right) \right],
$$

where $H(\mathbf{x})$ is a constant and we take $p(i) = \frac{1}{n}$ and $p(\mathbf{z}|i) = p(\mathbf{z}|\mathbf{x}_i)$. When the dataset containing $n$ independent and identically distributed (i.i.d) samples is large enough, the expectation of $p(\mathbf{x})$ can be regarded as uniform sampling from the dataset to get $\mathbf{x}_i$. Note that the lower bound is essentially identical to $\beta$VAE. The former represents the data reconstruction process and the latter is a regularization term for forgotten identity information, driving the acquisition of informative and generative representations, respectively.

## Appendix B. Derivation of Equation (8)

Same as Equation 2, we substitute the intractable posterior $p(\mathcal{X}|\mathbf{z})$ and marginal $p(\mathbf{z}^{(v)})$ with respective variational posterior $q(\mathcal{X}|\mathbf{z})$ and marginal $q(\mathbf{z}^{(v)})$, and then the variational lower bound for the first two terms of equation 8 yields as follows,

$$
\max_{\mathbf{Z}, \mathcal{Z}} I(\mathbf{Z}; \mathcal{X}) - \sum_v^m I(\mathbf{Z}^{(v)}; \mathbf{X}^{(v)})
$$

$$
= \int \int p(\mathbf{z}, \mathcal{X}) \log \frac{p(\mathcal{X}|\mathbf{z})}{p(\mathcal{X})} d\mathcal{X} d\mathbf{z} - \sum_m^v \int \int p(\mathbf{z}^{(v)}, \mathbf{x}^{(v)}) \log \frac{p(\mathbf{z}^{(v)}|\mathbf{x}^{(v)})}{p(\mathbf{z}^{(v)})} d\mathbf{x}^{(v)} d\mathbf{z}^{(v)}
$$

$$
= H(\mathcal{X}) + \mathbb{E}_{p(\mathcal{X})} \left[ \int p(\mathbf{z}|\mathcal{X}) \log p(\mathcal{X}|\mathbf{z}) d\mathbf{z} \right] - \sum_m^v \mathbb{E}_{p(\mathbf{x}^{(v)})} \left[ \int p(\mathbf{z}^{(v)}|\mathbf{x}^{(v)}) \log \frac{p(\mathbf{z}^{(v)}|\mathbf{x}^{(v)})}{p(\mathbf{z}^{(v)})} d\mathbf{z}^{(v)} \right]
$$

$$
\geq \mathbb{E}_{p(\mathcal{X})} \left[ \int p(\mathbf{z}|\mathcal{X}) \log q(\mathcal{X}|\mathbf{z}) d\mathbf{z} \right] - \sum_m^v \mathbb{E}_{p(\mathbf{x}^{(v)})} \left[ \int p(\mathbf{z}^{(v)}|\mathbf{x}^{(v)}) \log \frac{p(\mathbf{z}^{(v)}|\mathbf{x}^{(v)})}{q(\mathbf{z}^{(v)})} d\mathbf{z}^{(v)} \right]
$$

$$
= \mathbb{E}_{p(\mathcal{X})} \left[ \mathbb{E}_{p(\mathbf{z}|\mathcal{X})} \left[ \log q(\mathcal{X}|\mathbf{z}) \right] - \sum_v^m D_{KL}(p(\mathbf{z}^{(v)}|\mathbf{x}^{(v)}) \| q(\mathbf{z}^{(v)})) \right],
$$

where $H(\mathcal{X})$ is a constant. Given the unified multi-view representation $\mathbf{z}$, multi-view datasets $\mathcal{X}$ can be viewed as conditionally independent, i.e., $p(\mathcal{X}) = p(\mathbf{x}^{(1)}) p(\mathbf{x}^{(2)}) ... p(\mathbf{x}^{(m)})$. The first term maximizes the expectation of the log-likelihood to achieve comprehensive so that the multi-view representation retain more information of the data. The second term makes the multi-view representation forget the information of the original data by minimizing the KL divergence of the posterior distribution and the variational distribution to meet the requirement of concentration.

