# OpenReview forum: "Generalized Information-theoretic Multi-view Clustering"
_NeurIPS.cc/2023/Conference — NeurIPS 2023 poster_

### Official Review · Reviewer_Ayjx · 2023-06-21

**Soundness:** 3 good
**Presentation:** 2 fair
**Contribution:** 3 good
**Rating:** 6
**Confidence:** 4

**Summary:**

This paper proposes a new framework for unsupervised multi-view learning based on information bottleneck theory. The paper defines three desiderata for multi-view representation learning in terms of mutual information, namely, comprehensiveness, concentrate, and cross-diversity. The paper further introduces a clustering term to preserve the original data structure and leverages deep neural networks and stochastic gradient variational Bayes to optimize the objective function. The paper evaluates the proposed method on four real-world datasets and shows that it outperforms several state-of-the-art algorithms in terms of clustering performance.

**Strengths:**

•	This paper provides a general and principled information-theoretic framework for multi-view clustering that does not rely on strict assumptions about semantic consistency across views.

•	It incorporates three requirements for multi-view representation learning that balance the trade-off between informativeness, compression, and diversity of the latent features.


**Weaknesses:**

•	The authors should clarify how their definition of comprehensive, concentrative, and cross-diverse multi-view representation differs from the one used by Completer [22], which also maximizes the mutual information between views and minimizes conditional entropy of different views.

•	The loss function consists of four terms with different roles: data reconstruction, multi-regularization, information shift, and clustering. The authors should conduct ablation studies to show the contribution and necessity of each term for the proposed method.

•	The balance parameters $\beta$ and $\gamma$ control the trade-off in the objective function. The authors should provide some theoretical or empirical guidance for choosing the optimal values of these parameters for different datasets or scenarios.

•	As the robustness of the model is important to evaluate, what is the definition of robustness in this paper and the relation between the robustness and the proposed information bottleneck theory?

•	In Eq.(4), what is the difference between $\mathbf{Z}^{(v)}$ and $\mathbf{Z}$? How to learn $\mathbf{Z}$?


**Questions:**

The authors are encouraged to address the weaknesses pointed out in the previous section, such as conducting ablation studies. These improvements would make the contribution of the paper more evident and might increase my rating.

---

> ### Author Rebuttal · Authors · 2023-08-09
>
> We appreciate your recognition of our work and constructive comments.
>
> **Q1:** The authors should clarify how their definition of comprehensive, concentrative, and cross-diverse multi-view representation differs from the one used by Completer [22], which also maximizes the mutual information between views and minimizes conditional entropy of different views.
>
> **A1:** Completer maximizes the mutual information between views and minimizes the conditional entropy across view to learn the shared information between views, thus forcing the latent representations of different views to be consistent. The shared information is assumed to contain all task-related information from the data. However, this is not a reasonable assumption in practice, e.g., different views share the same background information but depict different foreground information.
>
> In contrast, the proposed model (IMC) is based on a general assumption that task-related information exists both in the view-shared part and in the view-peculiar part. Under the guidance of the information bottleneck principle, we seek to retain more intrinsic information (comprehensive) and compress each view as much as possible (concentrative), so we need to make each view contain various information (cross-diverse), respectively corresponding to three different mutual information. This is the motivation for proposing IMC, which is different from Completer.
>
> **Q2:** The loss function consists of four terms with different roles: data reconstruction, multi-regularization, information shift, and clustering. The authors should conduct ablation studies to show the contribution and necessity of each term for the proposed method.
>
> **A2:** I'm sorry that I did the ablation experiment and analysis, but I didn't set up a chapter on ablation study in the manuscript, which made it easy to be ignored. We will highlight the ablation study more clearly in the new version.
>
> In line 242 of Section 5.1 Experimental Settings, we construct two variants IMC-v1 and IMC-v2 to compare with IMC, and showed the experimental results in Table 1 and Table 2. The performance of using KL divergence clustering term has been significantly improved, while the use of the information shift term has a small improvement. And in Section 5.2 Performance Analysis, we concluded that end-to-end models help to learn more discriminative representations.
>
> **Q3:** The balance parameters and control the trade-off in the objective function. The authors should provide some theoretical or empirical guidance for choosing the optimal values of these parameters for different datasets or scenarios.
>
> **A3:** You point out one of the main limitation of this approach, we claim in chapter 5.4 that empirical guidance can be based on prior knowledge of the dataset. In our experiences, if there is a large degree of consistency in the semantics of multiple views, then the β is set to small and vice versa. As we mentioned in Chapter 6 there are practical limitations on parameter selection. Theoretical guidance for choosing the optimal parameters is a direction that can be researched in the future.
>
> **Q4:** As the robustness of the model is important to evaluate, what is the definition of robustness in this paper and the relation between the robustness and the proposed information bottleneck theory?
>
> **A4:** We believe that the robustness of the model is its ability to adapt to data generated in various complex scenarios, including noisy data. To evaluate the robustness of the proposed method, we added sparse noise of different scales to the dataset and the clustering results are shown in Table 2. It can be observed that the decline of IMC is the smallest, while the decline of the Completer model is obvious. This may be due to the assumption of multi-view semantic consistency is more sensitive as the noise ratio increases, the information shared by multiple views will also incorporate more noise.
>
> We are looking forward to adding other indicators to measure model robustness. If you have any suggestions on robustness metrics, it will greatly improve the quality of our work.
>
> **Q5:**  In Eq.(4), what is the difference between $\textbf{Z}^{(v)}$ and $\textbf{Z}$? How to learn $\textbf{Z}$?
>
> **A5:** $\textbf{Z}$ is the unified latent representation and$\textbf{Z}^{(v)}$ is the latent representation of the $v$-th view. We seek to learn a uniform multi-view representation that satisfies three desiderata, so we introduce the representations for each view. $\textbf{Z}$ can be obtained by the posterior $p(\textbf{z}|\textbf{z}^{(1)}, \textbf{z}^{(2)}, …, \textbf{z}^{(v)})$ which is modeled by a multiple fully connected layer (line 166). Intuitive understanding can be seen in Figure 2(c).

---

> > ### Comment · Reviewer_Ayjx · 2023-08-17
> >
> > After thoroughly reviewing the feedback from both the fellow reviewer and the author's rebuttal, I am of the opinion that this paper presents a new theory definition and employs a straightforward objective function to achieve cross-view diversity. Given these observations, I am inclined to maintain my current rating.

---

> > > ### Author Response · Authors · 2023-08-17
> > >
> > > We greatly appreciate your recognition and highlighting of the contributions to our work. Your praise is our passion for continuous improvement and motivation to contribute to the field.

---

### Official Review · Reviewer_bG4n · 2023-06-29

**Soundness:** 3 good
**Presentation:** 3 good
**Contribution:** 3 good
**Rating:** 6
**Confidence:** 4

**Summary:**

This paper reformulates the multi-view clustering problem from an information-theoretic perspective and propose a general theoretical framework. The authors extend the information bottleneck theory to unsupervised multi-view learning and achieve representation learning and clustering by leveraging deep neural networks and stochastic gradient variational Bayes.

**Strengths:**

1. This paper combines information theory with multi-view clustering and gives some new definitions to portray some properties of the multi-view domain, which is a very innovative idea that can contribute to the field.
2. This paper provides a solid theory and some key proofs are detailed and complete.


**Weaknesses:**

1. The datasets used in the experimental part are a bit less and small, and there are many challenging and large datasets in the field of multi-view clustering, the authors should add more experiments to enhance the sufficiency.
2. Parameter analysis part should show parameter changes on all datasets, you may show the figures of all datasets under one metric.

**Questions:**

1.in line 113, should the $I(Z^{(1)};X^{(2)})$ be $I(Z^{(2)};X^{(2)})$?

**Limitations:**

As the authors state in the summary section, the model is limited by the choice of parameters, and it would be a great improvement if a mathematical strategy could be found.

---

> ### Author Rebuttal · Authors · 2023-08-09
>
> Thanks for your compliment and constructive suggestion.
>
> **Q1:** The datasets used in the experimental part are a bit less and small, and there are many challenging and large datasets in the field of multi-view clustering, the authors should add more experiments to enhance the sufficiency.
>
> **A1:** Following your suggestion, we further test our model on a large-scale multi-view dataset.
>
> **NUS-WIDE-Object (NUS)** is a dataset for object recognition that consists of 30000 images in 31 classes. We use 5 features provided by the website, i.e. 65 dimension color Histogram (CH), 226 dimension color moments (CM), 145 dimension color correlation (CORR), 74 dimension edge distribution, and 129 wavelet texture.
>
> |  Datasets  |  Metrics  |  DMVAE | MIB | CMIB-Nets | Completer | IMC-v1 | IMC-v2 | IMC |
> |  ----  | ----  |  ----  | ----  |  ----  | ----  |  ----  | ----  | ----  |
> |  | ACC |   18.24$\pm$0.64 |   15.24$\pm$0.56 |   14.25$\pm$0.35 |   16.88$\pm$0.46 |   17.60$\pm$0.40 |   14.38$\pm$0.36 |   $\textbf{19.78} \pm \textbf{0.47}$ |
> | NUS  | NMI |  19.87$\pm$1.05 |   15.75$\pm$0.42 |   14.65$\pm$0.40 |  17.68$\pm$0.35 |   18.67$\pm$0.62 |   15.55$\pm$0.74 |   $\textbf{21.12} \pm \textbf{0.34}$ |
> |  | ARI |  6.78$\pm$0.81 |   5.83$\pm$0.84 |  4.26$\pm$0.74 |   6.24$\pm$0.64 |   5.68$\pm$0.48 |  5.68$\pm$0.48 |   $\textbf{8.32} \pm \textbf{0.41}$ |
> |
>
> From the results, it can be seen that NUS is a challenging dataset, and there is still much room for improvement in the clustering performance of all the models The optimal and sub-optimal performance achieved by the proposed IMC and DMVAE model demonstrates that incorporating the clustering loss to bootstrap the representation learning can significantly improve the clustering performance, which can be consistently concluded from the ablation experiments comparing IMC, IMC-v1, and IMC-v2.
>
> **Q2:** Parameter analysis part should show parameter changes on all datasets, you may show the figures of all datasets under one metric.
>
> **A2:** This suggestion is very constructive and I will post the parametric analysis of the NMI indicator for all datasets in the revised version, the results for the ACC and ARI indicators will be presented in the supplementary material.
>
> **Q3:** in line 113, should the $I(\textbf{Z}^{(1)}; \textbf{X}^{(2)})$ be $I(\textbf{Z}^{(2)}; \textbf{X}^{(2)})$?
>
> **A3:** I am sorry for the serious typos, here should be to minimize$I(\textbf{Z}^{(1)}; \textbf{X}^{(1)})$ and  $I(\textbf{Z}^{(2)}; \textbf{X}^{(2)})$, for maximum information compression of each view.

---

### Official Review · Reviewer_YKQa · 2023-07-01

**Soundness:** 2 fair
**Presentation:** 1 poor
**Contribution:** 2 fair
**Rating:** 6
**Confidence:** 5

**Summary:**

This paper presents an innovative information-theoretic framework for multi-view clustering, which overcomes the limitations of existing methods that rely on strict semantic consistency assumptions. By leveraging deep neural networks, the proposed method achieves more stable and superior clustering performance on several datasets.

**Strengths:**

The idea of incorporating the information bottleneck into multi-view clustering is intriguing and enlightening.

**Weaknesses:**

1. The importance of Eq. 3 in the proposed method is evident, as it provides the IB-based objective for the clustering approach. However, the paper lacks an explanation of how the original IB objective (Eq. 1) is transformed into the clustering objective, and why it takes the specific form presented. Additionally, if the information bottleneck based clustering is not an original contribution of this paper, it should be properly referenced.
2. The concept of the three desiderata is unclear. From Fig. 1, it appears that the final result of optimizing the three desiderata is to maximize I(Z; X^{1,2}), minimize I(Z; X{1,2}), and maximize I(Z; Z^{1,2}). This optimization seems contradictory, particularly when it aims to maximize the mutual information between Z and view-specific Z^{1,2} while minimizing that between raw data X and Z^{1,2}. Furthermore, the paper lacks an explanation of why these three desiderata are useful for learning cross-view representation, and there are no ablation studies to investigate their influence.
3. The proposed generalized multi-view clustering framework seems to differ from Eq. 3 only in the information shift term. This should be clarified and elaborated upon.
4. The experiments conducted on relatively simple and small-scale datasets limit the persuasiveness of the results. Additionally, the comparison baselines appear outdated, raising concerns about the effectiveness of the proposed method.
5. There are some typos, such as line 112: "I (Z^{(1)}; X^{(2)})."

**Questions:**

See weaknesses.

**Limitations:**

The authors have discussed the limitations of this paper.

---

> ### Author Rebuttal · Authors · 2023-08-09
>
> We greatly appreciate your thoughtful and detailed feedback.
>
> **A1:** Under the information bottleneck principle, [1] introduced the unsupervised information bottleneck objective Eq.(2), which is essentially the same as $\beta$VAE [2]. By summarizing the deep clustering methods DEC [3] and VaDE [4], we introduce the KL divergence term on the basis of Eq.(2) to discover the cluster structure of the data. The extent to which the clustering structure of the data is maintained while reducing dimensionality is the gap between Eq.(3) and Eq.(2).
>
> [1] Alexander A. Alemi, Ian Fischer, Joshua V. Dillon, and Kevin Murphy. Deep variational information bottleneck. In *ICLR*, 2017.
>
> [2] Irina Higgins, Loic Matthey, Arka Pal, Christopher Burgess, Xavier Glorot, Matthew Botvinick, Shakir Mohamed, and Alexander Lerchner. beta-vae: Learning basic visual concepts with a constrained variational framework. In *ICLR*, 2017.
>
> [3] Xie, Junyuan, Ross Girshick, and Ali Farhadi. Unsupervised deep embedding for clustering. In *ICML*, pages 478-487, 2016.
>
> [4] Zhuxi Jiang, Yin Zheng, Huachun Tan, Bangsheng Tang, and Hanning Zhou. Variational deep embedding: An unsupervised and generative approach to clustering. In *IJCAI*, 2017.
>
> **A2:** Actually, the optimization is a trade-off. Like the unsupervised information bottleneck theory ($\beta$VAE), it is desired to retain information about the data distribution ($\max I(\textbf{Z}; \textbf{X})$) and forget the information for identity samples ($\min I(\textbf{Z}; i)$). For the multi-view scenario, we seek to retain the multi-view data distribution information ($\max I(\textbf{Z}; \textbf{X}^{(1)},\textbf{X}^{(2)})$), and want to forget the single-view information ($\min I(\textbf{Z}^{(v)}, \textbf{X}^{(v)})$), and in order to match $\textbf{Z}$ and $\textbf{Z}^{(v)}$, we propose $\max I(\textbf{Z}; \textbf{Z}^{(1)}, \textbf{Z}^{(2)})$ as a trade-off glue (related proof in Proposition 3.1). There are two cases, when the information entropy of $\textbf{Z}$ is fixed, maximizing $I(\textbf{Z};\textbf{Z}^{(1)}, \textbf{Z}^{(2)})$ will increase the information entropy of $\textbf{Z}^{(1)}$ and $\textbf{Z}^{(2)}$. When the information entropy of $\textbf{Z}^{(1)}$, and $\textbf{Z}^{(2)}$ is fixed, maximizing $I(\textbf{Z};\textbf{Z}^{(1)}, \textbf{Z}^{(2)})$  will reduce the information entropy of $\textbf{Z}$. However, both cases are learning cross-view diversity.
>
> **For ablation study**, in line 242 of Section 5.1 Experimental Settings, we construct two variants IMC-v1 and IMC-v2 to compare with IMC, and showed the experimental results in Table 1 and Table 2. The performance of using KL divergence clustering term has been significantly improved, while the use of information shift term has a small improvement. And in Section 5.2 Performance Analysis, we concluded that end-to-end models help to learn more discriminative representations.
>
>
> **A3:** Eq. (3) is clustering based on single-view information bottleneck, while the proposed information-theoretic multi-view clustering (IMC) is for multi-view data. The maximum information shift term is used to connect the unified multi-view representation $\textbf{Z}$ and the view-specific representation $\textbf{Z}^{(v)}$ to achieve cross-diversity.
>
>
> **A4:** Following your suggestion, we further test our model on a large-scale multi-view dataset.
>
> **NUS-WIDE-Object (NUS)** is a dataset for object recognition that consists of 30000 images in 31 classes. We use 5 features provided by the web-site, i.e. 65 dimension color Histogram (CH), 226 dimension color moments (CM), 145 dimension color correlation (CORR), 74 dimension edge distribution, and 129 wavelet texture.
>
> |  Datasets  |  Metrics  |  DMVAE | MIB | CMIB-Nets | Completer | IMC-v1 | IMC-v2 | IMC |
> |  ----  | ----  |  ----  | ----  |  ----  | ----  |  ----  | ----  | ----  |
> |  | ACC |   18.24$\pm$0.64 |   15.24$\pm$0.56 |   14.25$\pm$0.35 |   16.88$\pm$0.46 |   17.60$\pm$0.40 |   14.38$\pm$0.36 |   $\textbf{19.78} \pm \textbf{0.47}$ |
> | NUS  | NMI |  19.87$\pm$1.05 |   15.75$\pm$0.42 |   14.65$\pm$0.40 |  17.68$\pm$0.35 |   18.67$\pm$0.62 |   15.55$\pm$0.74 |   $\textbf{21.12} \pm \textbf{0.34}$ |
> |  | ARI |  6.78$\pm$0.81 |   5.83$\pm$0.84 |  4.26$\pm$0.74 |   6.24$\pm$0.64 |   5.68$\pm$0.48 |  5.68$\pm$0.48 |   $\textbf{8.32} \pm \textbf{0.41}$ |
> |
>
> From the results, it can be seen that NUS is a challenging dataset, and there is still much room for improvement in the clustering performance of all the models The optimal and sub-optimal performance achieved by the proposed IMC and DMVAE model demonstrates that incorporating the clustering loss to bootstrap the representation learning can significantly improve the clustering performance, which can be consistently concluded from the ablation experiments comparing IMC, IMC-v1, and IMC-v2 .
>
> **A5:** I am sorry for the serious typos, here should be to minimize $I(\textbf{Z}^{(1)}; \textbf{X}^{(1)})$ and $I(\textbf{Z}^{(2)}; \textbf{X}^{(2)})$, for maximum information compression of each view.

---

> > ### Comment · Reviewer_YKQa · 2023-08-17
> >
> > I am pleased to acknowledge that the authors have effectively addressed my concerns. Their response has significantly clarified the contribution and technical details of the work. Additionally, they have incorporated additional experiments for a more comprehensive evaluation. Considering these improvements, I decided to revise my rating from "Borderline reject" to "Weak Accept".

---

> > > ### Author Response · Authors · 2023-08-17
> > >
> > > Thanks for taking your valuable time to read and respond in a timely manner. Your constructive comments contribute to the improvement of our work.

---

### Official Review · Reviewer_i2fg · 2023-07-06

**Soundness:** 2 fair
**Presentation:** 1 poor
**Contribution:** 2 fair
**Rating:** 4
**Confidence:** 4

**Summary:**

In this paper, the authors introduce representation learning with the unsupervised information bottleneck to multi-view clustering. Based on the framework of information bottleneck, the authors theoretically summarize 3 key properties (comprehensiveness, concentrate, and cross-diversity) required by multi-view clustering representation. Finally, a DEC module is added to obtain the clustering assignment.

**Strengths:**

- The paper is well-motivated and easy to follow. The pointed three desiderata are convincing although there may be some typos in the definition.
- The paper is technically sound.

**Weaknesses:**

- There are many typos, especially in key concepts. For example,
    - there may be plenty of typos of notations in Definition 3.1, which causes my major concern about the soundness of this paper.
    - In Line112-113, "minimize $I(Z^{(1)}, X^{(2)})$ and $I(Z^{(1)}, X^{(2)})$" seems wrong.
- The experiment may be a little insufficient. For example,
    - The ablation experiments to study which 3 different desiderata are more important are missing.
    - The visualization of learned $Z$ is also helpful to improve the quality. Since the primary contribution is the representation learning and there seems no apparent contribution to the clustering module, it is important to show whether the quality of representation is better.
    - The running time is missing.
- The used mathematical techniques to derive the variational bound lack novelty (widely used in VIB, GIB, etc.).

**Questions:**

- (The concept of comprehensiveness) In Definition 3.1, the definition of comprehensive seems wrong: " can be predicted by $Z$".
- It may be somewhat confusing to connect $\min I(Z, X)$ and "eliminating redundant information of each view". It should minimize the irrelevant information between $X$ and $Z$.
- In Line 116, why is Eq. 4 used for *comprehensiveness* and *concentrate*? Is it the proposed cross-diversity?

Overall, the paper has some merits but requires significant proofreading. I believe the paper could be better but the quality of this version is unsatisfactory. I'd like to update my score after reading the response and other reviews.

**Limitations:**

The limitations are not discussed in the main paper.

---

> ### Author Rebuttal · Authors · 2023-08-09
>
> We greatly appreciate your thoughtful and constructive feedback.
>
> **Q1:** In Line112-113, "minimize and $I(\textbf{Z}^{(1)}; \textbf{X}^{(2)})$ and $I(\textbf{Z}^{(1)}; \textbf{X}^{(2)})$" seems wrong.
>
> **A1:** I am sorry for the serious typos, here should be to minimize $I(\textbf{Z}^{(1)}; \textbf{X}^{(1)})$ and $I(\textbf{Z}^{(2)}; \textbf{X}^{(2)})$, for maximum information compression of each view.
>
> **Q2:** The ablation experiments to study which 3 different desiderata are more important are missing.
>
> **A2:** I'm sorry that I did the ablation experiment and analysis, but I didn't set up a chapter on ablation study in the manuscript, which made it easy to be ignored. We will highlight the ablation study more clearly in the new version.
>
> In line 242 of Section 5.1 Experimental Settings, we construct two variants IMC-v1 and IMC-v2 to compare with IMC, and showed the experimental results in Table 1 and Table 2. The performance of using KL divergence clustering term has been significantly improved, while the use of information shift term has a small improvement. And in Section 5.2 Performance Analysis, we concluded that end-to-end models help to learn more discriminative representations.
>
> **Q3:** The visualization of learned $\textbf{Z}$ is also helpful to improve the quality. Since the primary contribution is the representation learning and there seems no apparent contribution to the clustering module, it is important to show whether the quality of representation is better.
>
> **A3:** Thanks for your constructive suggestions. The representation visualization of multi-view learning can intuitively see the latent space structure after data dimensionality reduction, which is usually strongly related to the clustering metrics. We visualize the latent representations of all datasets and compare them with other multi-view representation learning methods, and we will present the results in the supplementary material due to the page limit of the main text.
>
> **Q4:** The running time is missing.
>
> **A4:** We tested the running time of 10 runs for 20 epochs on the UCI-digits dataset on a computer with an NVIDIA RTX 2070 GPU. The results show that Completer has the fastest running time, followed by our proposed model, which is close to MIB, because the multidimensional Jensen-Shannon estimator with high computational complexity is used.
>
> |   | DMVAE  | MIB  | Completer | IMC |
> |  ----  | ----  | ----  | ----  | ----  |
> | Running Time /s | 12.58 $\pm$ 0.12 | 8.41 $\pm$ 0.15 | 3.65 $\pm$ 0.20 | 7.20 $\pm$ 0.12|
> |
>
> The results show that Completer has the fastest running time, followed by our proposed model, which is close to MIB, because the multivariate Jensen-Shannon estimator with high computational complexity is used.
>
> **Q5:** (The concept of comprehensiveness) In Definition 3.1, the definition of comprehensive seems wrong: " can be predicted by
> $\textbf{Z}$".
>
> **A5:** There may be a bit of ambiguity here. The comprehensiveness means that multi-view observations $ \textbf{X}^{(1)}, \textbf{X}^{(2)}, ..., \textbf{X}^{(V)}$ can be predicted/generated by the unified representation $\textbf{Z}$.
>
> **Q6:** It may be somewhat confusing to connect $\min I(\textbf{Z}; \textbf{X})$ and "eliminating redundant information of each view". It should minimize the irrelevant information between $\textbf{X}$ and $\textbf{Z}$.
>
> **A6:** In the unsupervised setting, without labels, it's not known which information in the data is irrelevant to the task. It is desired to keep the principal and intrinsic information about the sample and eliminate redundant information such as background and noise.
>
> **Q7:** In Line 116, why is Eq. 4 used for comprehensiveness and concentrate? Is it the proposed cross-diversity?
>
> **A7:** Like the unsupervised information bottleneck theory ($\beta$VAE), it is desired to retain information about the data distribution ($\max I(\textbf{Z}; \textbf{X})$) and forget the information for identity samples ($\min I(\textbf{Z}; i)$). For the multi-view scenario, we seek to retain the multi-view data distribution information ($\max I(\textbf{Z}; \textbf{X}^{(1)},\textbf{X}^{(2)})$), and want to forget the single-view information ($\min I(\textbf{Z}^{(v)}, \textbf{X}^{(v)})$), and in order to match $\textbf{Z}$ and $\textbf{Z}^{(v)}$, we propose $\max I(\textbf{Z}; \textbf{Z}^{(1)}, \textbf{Z}^{(2)})$ as a trade-off glue (related proof in Proposition 3.1). There are two cases, when the information entropy of $\textbf{Z}$ is fixed, maximizing $I(\textbf{Z};\textbf{Z}^{(1)}, \textbf{Z}^{(2)})$ will increase the information entropy of $\textbf{Z}^{(1)}$ and $\textbf{Z}^{(2)}$. When the information entropy of $\textbf{Z}^{(1)}$, and $\textbf{Z}^{(2)}$ is fixed, maximizing $I(\textbf{Z};\textbf{Z}^{(1)}, \textbf{Z}^{(2)})$  will reduce the information entropy of $\textbf{Z}$. However, both cases are learning cross-view diversity.

---

> ### Comment · Reviewer_i2fg · 2023-08-17
>
> I thank the authors for the response and the review partially addressed my conerns (such as time and crucial formulation errors).
>
> I have read other reviews as well. I may not agree with that the paper present a new theory for multi-view clustering. The information bottleneck has been studied well and it is also applied to the unsupervised scenes. Moreover, the writing quality is poor so it may need to be polished more carefully before the formal publication.
>
> So I retain my initial rating.

---

> > ### Author Response · Authors · 2023-08-17
> >
> > First of all, thanks for your professional and pertinent comments, we try to communicate amicably to minimize the comprehension gap, and finally reach a consensus!
> >
> > As presented in the chapter "Related Work", the information bottleneck principle has been extensively studied, as has recent research on multi-view clustering. Representative works include MIB [1] and Completer [2], but they are all based on a strict assumption that multi-view shared information is learned as task-relevant information.
> >
> > The Information-theoretic Multi-view Clustering (IMC) approach proposed in this paper is based on the general assumption of multi-view data that task-relevant information exists in both view-sharing and view-specific parts. Theoretical analysis shows that the introduction of novel information shift term enables cross-view diversity, and extended experiments demonstrate that IMC is more robust both on various multi-view datasets and on noisy datasets.
> >
> > We will carry out professional and native language polishing of the manuscript before the final version, to resolve misunderstandings caused by improper expression.
> >
> > We are keen to further solve the remaining issues after rebuttal,  and look forward to your follow-up constructive comments to help us improve this work！
> >
> > [1] Marco Federici, Anjan Dutta, Patrick Forré, Nate Kushman, and Zeynep Akata. Learning robust representations via multi-view information bottleneck. In *ICLR*, 2021.
> >
> > [2] Yijie Lin, Yuanbiao Gou, Xiaotian Liu, Jinfeng Bai, Jiancheng Lv, and Xi Peng. Dual contrastive prediction for incomplete multi-view representation learning. *IEEE Trans. Pattern Anal. Mach. Intell.*, 2022.

---

### Comment · Area_Chair_UJz4 · 2023-08-13
**Post-rebuttal review**

Dear Reviewers,

Thanks for your time and efforts in reviewing this paper. The author rebuttal is now available. Please read the rebuttal and update your comments ASAP. Thank you very much!

Best,
AC

---

### Decision · Program_Chairs · 2023-09-21

**Decision:**

Accept (poster)

**Comment:**

This paper proposes a general multi-view clustering framework based on the information bottleneck principle [9], and put the framework into effect via utilizing deep neural networks and stochastic gradient variational Bayes. After the author rebuttal, the majority of the reviews agree that the paper is contributive and suggest acceptance. Reviewer i2fg acknowledges the author rebuttal that has partially addressed their concerns, but continues complaining the novelty of the proposed theories. Based on the author-review communications, the review-review discussions, and my own read, my assertion is that the technical novelty is significant enough, though the presented framework and theories are not completely new. Thus I would suggest accepting the paper.